# Graph Low-rank Non-negative Matrix Factorization with Auto-encoders for Fault Detection

1st Yabing Liu
*School of Electronic and Information Engineering*
*Southwest University*
Chongqing, China
liuyabing@email.swu.edu.cn

2nd Shanshan Yu*
*School of Electronic and Information Engineering*
*Southwest University*
Chongqing, China
*Key Laboratory of Cyber-Physical Fusion*
*Intelligent Computing*
*South-Central Minzu University*
*State Ethnic Affairs Commission*
Wuhan 430074, China
yu33@swu.edu.cn

3rd Wei Guo
*School of Electronic and Information Engineering*
*Southwest University*
Chongqing, China
guowei123@email.swu.edu.cn

4th Man-Fai Leung
*School of Computing and Information Science,*
Faculty of Science and Engineering
*Anglia Ruskin University*
Cambridge, United Kingdom
man-fai.leung@aru.ac.uk

*Abstract*—**Fault detection is the process of detecting and diagnosing faults or abnormalities in a system by analyzing its operational data. However, with the complexity of modern industrial processes, some faults are difficult to be detected in a timely manner due to various factors such as noise and data nonlinearity. Therefore, data-driven Fault Detection (FD) has become a widely used method to detect abnormal events in functional modules. Non-negative Matrix Factorization (NMF), as an efficient dimensionality reduction technique, has not had potential applications in fault detection (FD) thoroughly explored. In order to improve the FD methods based on NMF, we have developed a new approach, named graph low-rank non-negative matrix Factorization with auto-encoders (GLNMFA). GLNMFA integrates the Laplacian operator effectively identifies the local structure among data points, enhancing the performance of dimensionality reduction algorithms. It also introduces the nuclear norm to find a low-rank approximation to the original matrix, thereby constraining sparsity. Additionally, auto-encoders are incorporated to learn a low-dimensional representation of the data and extract key features, which are subsequently applied for fault detection purposes. We employ an optimization algorithm based on Alternating Direction Method of Multipliers (ADMM) to optimize this model. Two test statistics $T^2$ (Hotelling's T-squared), SPE (Squared Prediction Error) are used to evaluate detection efficiency. Kernel Density Estimation (KDE) are used to estimate control limits for fault detection. The effectiveness of GLNMFA is validated on two benchmark datasets.**

*Index Terms*—**Fault detection; Kernel density estimation; Nonnegative matrix factorization.**

## I. INTRODUCTION

In modern industrial production, it is essential to ensure the stable operation of equipment and the continuity of processes. The application of fault detection technology aims to discover and diagnose potential faults in time through real-time monitoring and analysis of various data in the production process, so as to take preventive measures to avoid production disruption. Fault detection methods can be divided into three categories: model-based method, signal-based method and data-driven method. Non-negative matrix factorization (NMF) has emerged as a powerful tool for data analysis, found applications across various domains. Its fundamental idea is to decompose a given data matrix into the product of non-negative basis vectors and coefficient matrices, enabling efficient representation and dimensionality reduction of the data. In recent years, NMF has shown significant potential in the field of fault detection. Fault detection is a critical task in industrial production and equipment maintenance [1]–[4]. Traditional methods for fault detection often rely on expert knowledge or statistical techniques, which may struggle to handle the

variability and high dimensionality of data in complex systems [5]. As a data-driven approach, NMF can automatically learn basic patterns from data, offering a new perspective and tool for fault detection [6]. Fault detection methods are generally divided into three kinds, which are signal driven, model driven and data driven. With the rapid development of data collection and data processing technology, data-driven fault detection has become the mainstream [7]. The most popular data-driven methods include Independent component analysis (ICA) [8], Principal Component Analysis (PCA) [9], canonical Component analysis (CCA) [10], and non-negative matrix factorization (NMF). NMF has been successfully applied in various applications, including feature extraction, topic modeling, and collaborative filtering. Its ability to discover underlying patterns in data and its interpretability make it a valuable tool in exploratory data analysis and dimensionality reduction tasks [11]. In the classical Non-negative Matrix Factorization (NMF) model, enhancing the model's performance can be achieved by incorporating regularization terms and constraints. This approach facilitates the extraction of more meaningful features and enhances the effectiveness of dimensionality reduction [12], [13]. For example, Sparse Non-negative Matrix Factorization (SNMF) produces more interpretive decomposition results by introducing sparsity constraints into the NMF model. This ensures that most elements in the generated base and coefficient matrices are zero [14]–[17]. Graph Regularized Non-negative Matrix Factorization (GNMF) introduces graph structure information based on standard NMF, and improves the quality and stability of decomposition results by using the relationship between data samples [18], [19], [20]. Orthogonal Non-negative Matrix Factorization (ONMF) introduces orthogonal constraints, i.e., the resulting basis matrix is orthogonal [17], [21] . The orthogonal constraint makes the basis matrix generated by NMF more sparse and mutually exclusive, thus enhancing the interpretation and generalization ability of decomposition results. The statistical strategy based on NMF was first developed by Lee and Seung [22]. Li et al was the first to apply NMF fault detection in non-Gaussian processes [23]. Two statistical metrics are established for fault detection using the NMF method, namely the squared prediction error (SPE) and squared distance statistic ($T^2$), $T^2$ can effectively combine information from multiple variables to detect anomalies by monitoring changes in these variables and their correlations, and kernel density estimates (KDE) were used to estimate the control limits [24], [25]. The data on the benchmark Tennessee Eastman process show that the fault detection method based on NMF has better performance. Later, many new NMF variants appeared and proved to have better fault detection performance [26], [27].

In this paper, a novel fault detection approach is presented, which is based on Non-negative Matrix Factorization (NMF) and is named Graph Low-rank Non-negative Matrix Factorization with Auto-encoder (GLNMFA). This method leverages the graph Laplacian to incorporate the topological relationships between process variables, thereby enhancing the model's ability to utilize information from these variables. Additionally, the method utilizes the nuclear norm to ensure a low-rank matrix approximation, which efficiently reduces model redundancy. This advancement is supported by the cited literature [28]–[30]. Auto-encoders play a crucial role in this framework by mapping high-dimensional data into a lower-dimensional space, simplifying data representation and reducing complexity, while retaining essential information [31]. This unsupervised learning algorithm achieves this by encoding input data into a compact representation and then reconstructing it through a decoder to closely match the original input [32], [33]. Then the fault detection efficiency of this model is discussed [34]–[37]. Based on analysis of benchmark datasets such as the Tennessee Eastman Process (TEP) and XJTU-SY rolling bearings, this paper demonstrates the advantages and potential applications of GLNMFA in fault detection [38], [39]. This is expected to improve the accuracy and efficiency of fault detection, providing more reliable support for industrial production and equipment maintenance. The remainder of the paper is outlined below, with Section II reviewing the classic NMF and some representative variants. Section III designs an effective algorithm to solve GLNMFA. Section IV introduces apply the GLNMFA model proposed in this paper for fault detection, Section V highlights the advantages of this algorithm by comparing with other methods, and finally, provides a summary of the paper.

## II. RELATED WORKS

### A. Notation

For a matrix $\mathbf{X} \in \mathbb{R}^{m \times N}$, where $m$ represents the number of variables and $N$ represents the number of samples, the notation $x_i$ represents the $i_{th}$ row of matrix $\mathbf{X}$, and $x_{ij}$ represents the element at the $i_{th}$ row and $j_{th}$ column of matrix $\mathbf{X}$, $|| \cdot ||_F$ is the Frobenius norm of the matrix. $\| \mathbf{X} \|_*$ is the nuclear norm of the matrix, representing the sum of all the singular values of the matrix. $\mathbf{X}^T$ and $\mathbf{X}^{-1}$ are expressed as their transposed and inverse matrices, respectively. The inner product of two matrix can be given by the following formula : $\langle \mathbf{X}, \mathbf{Y} \rangle = \mathrm{tr}(\mathbf{X}^T \mathbf{Y}) = \sum_{i=1}^{m} \sum_{j=1}^{N} x_{ij} y_{ij}$.

### B. NMF Basics

Mathematically, NMF can be formulated as : $\mathbf{X} \approx \mathbf{WH}$. $\mathbf{X}$ is the original data matrix, $\mathbf{W} \in \mathbb{R}^{m \times k}$ contains

the basis vectors, called the basis matrix, and $\mathbf{H} \in \mathbb{R}^{k \times N}$ contains the coefficients, called the coefficient matrix, in which $k$ is the reduced dimension satisfying $(m + N) \times k < m \times N$. The loss function of NMF is usually defined as the distance between the original matrix $\mathbf{X}$ and the approximately reconstructed matrix $\mathbf{WH}$, and the difference between the two is usually measured using the Euclidean distance or the distance based on the KL divergence. According to Lee and Seung's research, the loss function can be reasonably defined as :

$$\min_{\mathbf{W},\mathbf{H}} \frac{1}{2} \parallel \mathbf{X} - \mathbf{WH} \parallel_F^2 \quad (1)$$
$$\text{s.t.} \mathbf{W} \geq 0, \mathbf{H} \geq 0.$$

In this case, $\mathbf{W}$ and $\mathbf{H}$ are greater than or equal to zero, which means that their elements are non-negative. There are many optimization methods for NMF-related problems, such as multiplication update (MU), gradient descent (PGD) [40], etc. But these algorithms converge too slowly. A new alternating direction multiplier method (ADMM) shows good convergence performance when solving NMF-related problems. In this paper, ADMM algorithm is used to solve the target problem.

Adding some constraints or regularization terms to the classical NMF model can improve the performance of the model, for example, graph regularized NMF (GNMF) [18].

$$\min_{\mathbf{W},\mathbf{H}} \frac{1}{2} \|\mathbf{X} - \mathbf{WH}\|_F^2 + \lambda \text{tr}(\mathbf{HLH}^{\text{T}}) \quad (2)$$
$$\text{s.t.} \mathbf{W} \geq 0, \mathbf{H} \geq 0.$$

In the model, a graph regularization term is added to the classical NMF, where $\mathbf{L}$ is referred to as the Laplacian matrix learned from the original matrix $\mathbf{X}$, defined as ($\mathbf{L} = \mathbf{D}$ - $\mathbf{Z}$). Here, $\mathbf{D}$ represents the adjacency matrix, and $\mathbf{Z}$ denotes the degree matrix. The addition of this regularization term can better capture the local structure information among the data points, thus improving the performance of data dimensionality reduction [41].

Another popular variant of NMF is obtained by placing a constraint on the coefficient matrix $\mathbf{H}$, named sparse nonnegative matrix decomposition (SNMF) [14].

$$\min_{\mathbf{W},\mathbf{H}} \frac{1}{2} \|\mathbf{X} - \mathbf{WH}\|_F^2 \quad (3)$$
$$\text{s.t.} \mathbf{W} \geq 0, \mathbf{H} \geq 0, \|\mathbf{H}\|_0 \leq s.$$

Compared with the classical NMF model, the above model adds additional constraints on the coefficient matrix $\mathbf{H}$. Here, the sparsity of $\mathbf{H}$ is controlled by the $l_0$ norm constraint on $\mathbf{H}$, so that $\mathbf{H} \leq s$ to achieve better dimensionality reduction. $s$ here is a parameter that controls the sparsity of the coefficient matrix H.

## III. GRAPH LOW-RANK NON-NEGATIVE MATRIX FACTORIZATION WITH AUTO-ENCODERS

### A. Model description

This section introduces the proposed NMF model. In order to obtain better performance, two regularization terms, the nuclear norm term and the auto-encoder term, are added on the basis of GNMF are added on the basis of GNMF, nuclear norm term and auto-encoder term. The nuclear norm regularization term can achieve the low-rank approximation of the matrix, thus achieving the effect of dimensionality reduction and denoising, auto-encoding terms can learn a low-dimensional representation of the data to extract key features.

$$\min_{\mathbf{W},\mathbf{H}} \frac{1}{2} \parallel \mathbf{X} - \mathbf{WH} \parallel_F^2 + \lambda_1 \parallel \mathbf{H} - \mathbf{W}^{\text{T}}\mathbf{X} \parallel_F^2 + \lambda_2 \parallel \mathbf{H} \parallel_*$$
$$+ \lambda_3 \text{tr}(\mathbf{HLH}^{\text{T}})$$
$$\text{s.t.} \mathbf{W} \geq 0, \mathbf{H} \geq 0.$$
$$(4)$$

$\parallel \mathbf{H} \parallel_*$ is the nuclear norm of $\mathbf{H}$, $\lambda_1, \lambda_2, \lambda_3$ is the regularization parameter. In fact, it can be viewed as an extension of Model (2).

### B. Optimization algorithm

The main idea of ADMM is to decompose the original problem into multiple sub-problems, and gradually approach the optimal solution of the original problem by optimizing each sub-problem alternatively [42], [43]. The optimization of nuclear norm is complicated. Meanwhile, in order to simplify the model optimization process, introduce two auxiliary variables $\mathbf{Y}$ and $\mathbf{U}$, then equation (4) can be reformulated as follows:

$$\min_{\mathbf{W},\mathbf{H},\mathbf{U},\mathbf{Y}} \frac{1}{2} \parallel \mathbf{X} - \mathbf{Y} \parallel_F^2 + \lambda_1 \parallel \mathbf{H} - \mathbf{W}^{\text{T}}\mathbf{X} \parallel_F^2$$
$$+ \lambda_2 \parallel \mathbf{U} \parallel_* + \lambda_3 \text{tr}(\mathbf{HLH}^{\text{T}}) \quad (5)$$
$$\text{s.t.} \mathbf{WH} = \mathbf{Y}, \mathbf{H} = \mathbf{U}, \mathbf{W} \geq 0, \mathbf{H} \geq 0.$$

Construct an augmented Lagrange of the original function as follows:

$$\mathcal{L}(\mathbf{W}, \mathbf{H}, \mathbf{U}, \mathbf{Y}, \mathbf{A}, \mathbf{B}) =$$
$$\frac{1}{2} \parallel \mathbf{X} - \mathbf{Y} \parallel_F^2 + \lambda_1 \parallel \mathbf{H} - \mathbf{W}^{\text{T}}\mathbf{X} \parallel_F^2$$
$$+ \lambda_2 \parallel \mathbf{U} \parallel_* + \lambda_3 \text{tr}(\mathbf{HLH}^{\text{T}})$$
$$+ \frac{\beta_1}{2} \|\mathbf{WH} - \mathbf{Y}\|_F^2 - \langle \mathbf{A}, \mathbf{WH} - \mathbf{Y} \rangle$$
$$+ \frac{\beta_2}{2} \|\mathbf{H} - \mathbf{U}\|_F^2 - \langle \mathbf{B}, \mathbf{H} - \mathbf{U} \rangle.$$
$$(6)$$

where $\beta_1, \beta_2$ are penalty parameters, and $\mathbf{A}$, $\mathbf{B}$ are Lagrange multipliers. Here also need to consider the non-negative constraints on $\mathbf{W}$ and $\mathbf{H}$. Under the framework of ADMM, the optimal solution is iteratively obtained

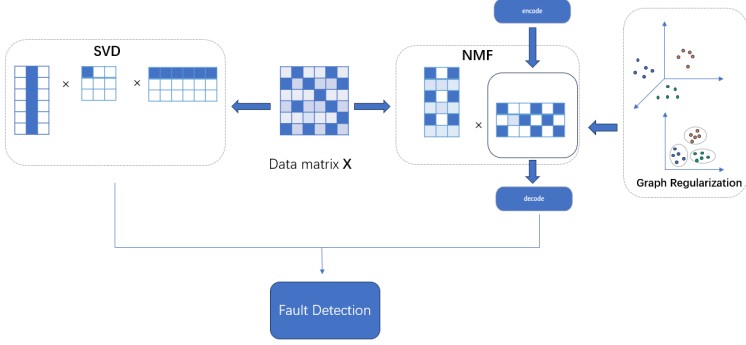

Fig. 1: The illustration of the proposed model.

by updating variables one by one [44]. The iterative steps for solving (6) are as follows:

$$\mathbf{W}_{k+1} = \arg\min_{\mathbf{W}} \mathcal{L}(\mathbf{W}, \mathbf{H}_k, \mathbf{U}_k, \mathbf{Y}_k, \mathbf{A}_k, \mathbf{B}_k)$$

$$\mathbf{H}_{k+1} = \arg\min_{\mathbf{H}} \mathcal{L}(\mathbf{W}_{k+1}, \mathbf{H}, \mathbf{U}_k, \mathbf{Y}_k, \mathbf{A}_k, \mathbf{B}_k)$$

$$\mathbf{U}_{k+1} = \arg\min_{\mathbf{U}} \mathcal{L}(\mathbf{W}_{k+1}, \mathbf{H}_{k+1}, \mathbf{U}, \mathbf{Y}_k, \mathbf{A}_k, \mathbf{B}_k)$$

$$\mathbf{Y}_{k+1} = \arg\min_{\mathbf{Y}} \mathcal{L}(\mathbf{W}_{k+1}, \mathbf{H}_{k+1}, \mathbf{U}_{k+1}, \mathbf{Y}, \mathbf{A}_k, \mathbf{B}_k)$$

$$\mathbf{A}_{k+1} = \mathbf{A}_k - \beta_1(\mathbf{W}_{k+1}\mathbf{H}_{k+1} - \mathbf{Y}_{k+1})$$

$$\mathbf{B}_{k+1} = \mathbf{B}_k - \beta_2(\mathbf{H}_{k+1} - \mathbf{U}_{k+1}).$$

The update process for each variable is discussed in detail below:

(1) Update $\mathbf{W}$ while fixing other variables.

updating $\mathbf{W}$ can be simplified as following optimization problems.

$$\min_{\mathbf{W}} \lambda_1 \parallel \mathbf{H} - \mathbf{W}^{\mathrm{T}}\mathbf{X} \parallel_F^2 + \frac{\beta_1}{2} \|\mathbf{W}\mathbf{H}_k - \mathbf{Y}_k - \mathbf{A}_k/\beta_1\|_F^2. \tag{7}$$

$$2\lambda_1\mathbf{X}\mathbf{X}^{\mathrm{T}}\mathbf{W}_{k+1} + \beta_1\mathbf{W}_{k+1}H_k\mathbf{H}_k^{\mathrm{T}} = \\ 2\lambda_1\mathbf{X}\mathbf{H}_k^{\mathrm{T}} + \beta_1\mathbf{Y}_k\mathbf{H}_k^{\mathrm{T}} + \mathbf{A}_k\mathbf{H}_k^{\mathrm{T}}. \tag{8}$$

Equation (8) is the result after derivation, which belongs to the Sylvester equation, If an equation form such as:

$$\mathbf{AX + XB = C}.$$

$\mathbf{A} \in \mathbb{R}^{m \times m}, \mathbf{B} \in \mathbb{R}^{n \times n}, \mathbf{X}$ and $\mathbf{B} \in \mathbb{R}^{m \times n}$, then this equation is called the Sylvester equation, Bartels et al. have given an introduction to solving this equation in detail, and this paper does not go into detail [45]. In equation (8), let $A = 2\lambda_1\mathbf{X}\mathbf{X}^{\mathrm{T}}$, $B = \beta_1 H_k\mathbf{H}_k^{\mathrm{T}}$, $C = 2\lambda_1\mathbf{X}\mathbf{H}_k^{\mathrm{T}} + \beta_1\mathbf{Y}_k\mathbf{H}_k^{\mathrm{T}} + \mathbf{A}_k\mathbf{H}_k^{\mathrm{T}}$, it is easy to see that the solution to $\mathbf{W}$ is solved in accordance with the form of the Sylvester matrix.

(2) Update $\mathbf{H}$ while fixing other variables. updating $\mathbf{H}$

can be simplified as following optimization problems.

$$\min_{\mathbf{H}} \lambda_1 \parallel \mathbf{H} - \mathbf{W}_{k+1}^{\mathrm{T}}\mathbf{X} \parallel_F^2 + \lambda_3\mathrm{tr}(\mathbf{HLH}^{\mathrm{T}})$$

$$+ \frac{\beta_1}{2}\|\mathbf{W_{k+1}H} - \mathbf{Y}_k\|_F^2 - \langle\mathbf{A_k}, \mathbf{W_{k+1}H} - \mathbf{Y_k}\rangle \tag{9}$$

$$+ \frac{\beta_2}{2}\|\mathbf{H} - \mathbf{U_k}\|_F^2 - \langle\mathbf{B_k}, \mathbf{H}_k - \mathbf{U}_k\rangle$$

$$(2\lambda_1\mathbf{I} + \beta_1\mathbf{W}_{k+1}^{\mathrm{T}}\mathbf{W}_{k+1} + \beta_2\mathbf{I})\mathbf{H}_{k+1} + \mathbf{H}_{k+1}(2\lambda_3\mathbf{L}) \\ = 2\beta_1\mathbf{W}_{k+1}^{\mathrm{T}}\mathbf{Y}_k + \mathbf{W}_{k+1}^{\mathrm{T}}\mathbf{A}_k + \lambda_1\mathbf{W}_{k+1}^{\mathrm{T}}\mathbf{X} + \beta_2\mathbf{U}_k + \mathbf{B}_k. \tag{10}$$

Equation (10) can also be seen as a Sylvester equation through simple calculation, which can also be solved. In this equation, $\mathbf{I}$ is the identity matrix , $\mathbf{I} \in \mathbb{R}^{k \times k}$.

(3) Update $\mathbf{Y}$ while fixing other variables. updating $\mathbf{Y}$ can be simplified as following optimization problems.

$$\min_{\mathbf{Y}} \frac{1}{2}\|\mathbf{X} - \mathbf{Y}\|_F^2 + \frac{\beta_1}{2}\|\mathbf{Y} - \mathbf{W}_{k+1}\mathbf{H}_{k+1} + \mathbf{A}_k/\beta_1\|_F^2. \tag{11}$$

The following solution is obtained through derivation:

$$\mathbf{Y}_{k+1} = \frac{1}{1 + \beta_1}(\mathbf{X} + \beta_1\mathbf{W}_{k+1}\mathbf{H}_{k+1} - \mathbf{A}_k). \tag{12}$$

(4) Update $\mathbf{U}$ while fixing other variables. updating $\mathbf{U}$ can be simplified as following optimization problems.

$$\min_{\mathbf{U}} \lambda_2\|\mathbf{U}\|_* - \frac{\beta_2}{2}\|\mathbf{H}_{k+1} - \mathbf{U} - \mathbf{B}_k/\beta_2\|_F^2. \tag{13}$$

Since the nuclear norm is not differentiable in most cases, and this form of formula has a closed solution, it is solved as follows:

a. Singular value decomposition (SVD) [46]: Decomposition of matrix $\mathbf{A}$ into the product of three matrices $\mathbf{U}$, $\Sigma$, $\mathbf{V}$. $\mathbf{A} = \mathbf{U}\Sigma\mathbf{V}^{\mathrm{T}}$, $\Sigma$ is a diagonal matrix containing singular values of matrix $\mathbf{A}$.

b. Select a threshold: Determine a threshold $\tau$, $\tau$ is used to determine the magnitude of the singular value.

c. Threshold processing: The elements of the $\Sigma$ matrix

is less than or equal to the threshold value The singular values of $\tau$ is set to 0, preserving singular values greater than the threshold.

$$\sigma'_i = \begin{cases} \sigma_i & \text{if} \quad \sigma_i > \tau \\ 0 & \text{if} \quad \sigma_i \leq \tau \end{cases}.$$

d. Reframe: using the processed $\Sigma$ matrix, and reconstruct $\mathbf{A}'$ matrix to $\mathbf{A}' = \mathbf{U}\Sigma'\mathbf{V}^T$.

e. $\mathbf{A}'$ is a simplified or denoised version of the original matrix $\mathbf{A}$.

---

**Algorithm 1 ADMM for GLNMFA**

---

**Input:** Given the original data matrix $\mathbf{X}$, Laplace matrix $\mathbf{L}$, parameter$\lambda_1, \lambda_2, \lambda_3 > 0$, penalty parameters $\beta_1, \beta_2 > 0$.
**Initialize:** $(\mathbf{W}_0, \mathbf{H}_0, \mathbf{U}_0, \mathbf{Y}_0, \mathbf{A}_0, \mathbf{B}_0)$, set k=0.
**Repeat:**
1: Update $\mathbf{W}_{k+1}$ by (8).
2: Update $\mathbf{H}_{k+1}$ by (10).
3: Update $\mathbf{Y}_{k+1}$ by (12).
4: Update $\mathbf{U}_{k+1}$ by (13).
5: Update $\mathbf{A}_{k+1} = \mathbf{A}_k - \beta_1(\mathbf{W}_{k+1}\mathbf{H}_{k+1} - \mathbf{Y}_{k+1})$.
6: Update $\mathbf{B}_{k+1} = \mathbf{B}_k - \beta_2(\mathbf{H}_{k+1} - \mathbf{U}_{k+1})$.
**End While**

---

## IV. APPLICATION STUDIES

This section introduces the fault detection process based on GLNMFA and conducts a comparative analysis with PCA, NMF, and SNMF using the TEP and XJTU-SY Bearing Dataset. The aim is to establish that GLNMFA demonstrates higher detection efficiency than other algorithms across the majority of fault variables. All experiments in this article were conducted on Windows 10, Intel(R) Core(TM) i7-8750H, CPU of 2.21 GHz, RAM of 16.0 GB, using Matlab R2020b.

frame diagram.png

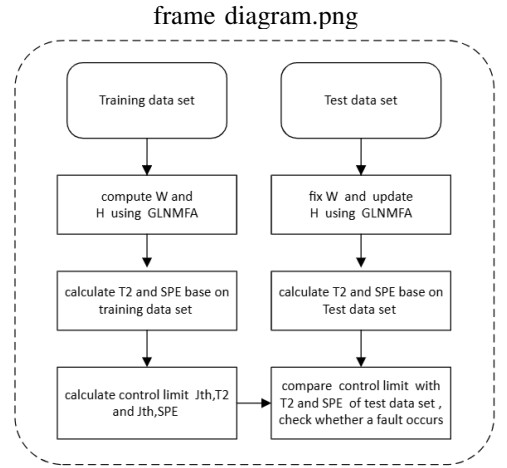

Fig. 2: Control frame diagram of fault detection.

### A. Fault Detection Process based on GLNMFA

Fig. 2 described the fault detection process based on Graph low-rank non-negative matrix decomposition with auto-encoding(GLNMFA-FD) method, detailed steps are as follows:

(1) Initialization: given the Normal Sample (not-fault) data matrix $\mathbf{X}$, builds the Laplace matrix $\mathbf{L}$ of the original matrix. $\mathbf{W}$ and $\mathbf{H}$ are initialized using positive random numbers.

(2) Calculate $\mathbf{W}$ and $\mathbf{H}$: The GLNMFA model is optimized by ADMM algorithm, and the decomposed radix matrix $\mathbf{W}$ and coefficient matrix $\mathbf{H}$ are obtained.

(3) Calculate $T^2$ and $SPE$: The coefficient matrix $\mathbf{H}$ can be viewed as a low-rank approximation of the original data matrix. As stated in [23], $\mathbf{H}$ reflects the state of the industrial process. To reconstruct $\mathbf{H}$ and $\mathbf{X}$, the formula is as follows:

$$\hat{\mathbf{H}} = (\mathbf{W}^{\mathrm{T}}\mathbf{W})^{-1}\mathbf{W}^{\mathrm{T}}\mathbf{X} \tag{14}$$

$$\hat{\mathbf{X}} = \mathbf{W}\hat{\mathbf{H}} \tag{15}$$

In the process of fault detection based on non-negative matrix factorization, two fault detection indexes $T^2$ and $SPE$ are constructed as follows:

$$\begin{aligned} T^2 &= \hat{\mathbf{H}}^{\mathrm{T}}\hat{\mathbf{H}} \\ \mathrm{SPE} &= (\mathbf{X} - \hat{\mathbf{X}})^{\mathrm{T}}(\mathbf{X} - \hat{\mathbf{X}}). \end{aligned} \tag{16}$$

(4) Computational control limit: The upper control limits for the two monitoring metrics $T^2$ and $SPE$ are calculated using KDE. The KDE equation for univariate kernel estimation is shown in equation (17):

$$\hat{P}(x) = \frac{1}{Nh}\sum_i K\left(\frac{x - x_i}{h}\right) \tag{17}$$

$\hat{P}(x)$ is the estimate of the probability density function, where N is the number of samples, $h$ is the bandwidth, and $K(\cdot)$ is the kernel function. The requirements for kernel functions are shown in equation (18).

$$\int_{-\infty}^{+\infty} K(x)\mathrm{d}x = 1, K(x) \geqslant 0 \tag{18}$$

The corresponding control limits for $T^2$ statistics and SPE statistics are denoted as $J_{th,T^2}$ and $J_{th,SPE}$, respectively.

(5) According to the test data, a new coefficient matrix $\mathbf{H}$ is obtained. New $T^2$ and $SPE$ are calculated according to equations (16).

(6) Determine whether the new $T^2$ and $SPE$ are faulty by comparing them with the corresponding control limits. If the test statistics are out of control, a fault occurs, otherwise normal. Therefore, the detection logic can be defined as:

$$\begin{cases} T^2 < J_{\mathrm{th},T^2} \text{ and } \mathrm{SPE} < J_{\mathrm{th,SPE}} \Rightarrow \mathrm{fault-free} \\ T^2 \geq J_{\mathrm{th},T^2} \text{ or } \mathrm{SPE} \geq J_{\mathrm{th,SPE}} \Rightarrow \mathrm{faulty}. \end{cases} \tag{19}$$

(7) Two percentage parameters, MAR (Miss alarm

rate) and FAR (False alarm rate), are used as criteria to judge fault detection. This formula means that the smaller MAR and FAR, the better the detection results.

$$\text{MAR} = \frac{\text{number of samples } (T^2 < J_{\text{th},T^2} \mid f \neq 0)}{\text{total of samples } (f \neq 0)}$$

$$\text{FAR} = \frac{\text{number of samples } (T^2 \geq J_{\text{th},T^2} \mid f = 0)}{\text{total of samples } (f = 0)}. \tag{20}$$

### B. Application on the TEP

The Benchmark Tennessee Eastman process is a standardized platform for testing process monitoring and fault diagnosis algorithms [47]. Fig. 3 shows the complex chemical process for TEP, including 22 processing units and 12 process variables [17]. TE process is highly nonlinear and multi-variable, so it is widely used in research for monitoring algorithms. Its standardization promotes the performance comparison of algorithms and the development of techniques. TE process data sets are used to evaluate the accuracy and reliability of algorithms, which is of great significance to improve the efficiency and safety of chemical processes. In the PCA based approach [48], the selection variance contribution is 85%, and the corresponding k value is 20 and the number of iterations is set to 1000.

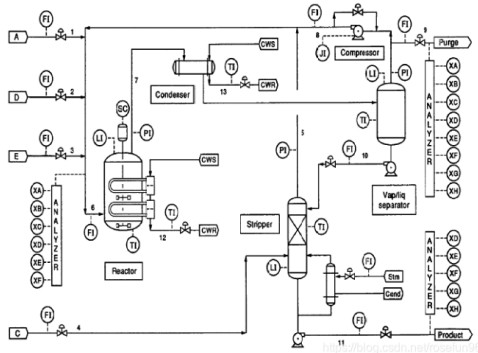

Fig. 3: Flowchart of the TEP.

The detection results based on PCA and NMF methods are shown in Table I and Table II, respectively. In addition, the best performances are highlighted in bold. In Table I and Table II, $T^2$, SPE represents the percentage of the fault variable that is greater than the control limit in the total sample. In the Fig. 5 to Fig. 9, $T^2$, SPE represents the actual value calculated according to the fault detection strategy. It can be seen from the data in the Table I-II that although GLNMFA is weaker than individual algorithms at individual fault points, GLNMFA has the best fault detection efficiency overall.

### C. Application on the XJTU-SY Bearing Dataset

The XJTU-SY rolling bearing acceleration life test dataset provided by Xi'an Jiaotong University is a valuable resource containing vibration signals and lifespan data of rolling bearings under different operating conditions [49]. It is primarily used for research in bearing life prediction and health monitoring, aiding in the development and evaluation of machine learning and deep learning models. Fig. 4 Experimental platform includes AC motor, motor speed controller, shaft, support bearing, hydraulic loading system and test bearing [50]. Data are collected using a portable dynamic signal acquisition unit at a sampling frequency of 25.6 kHz, with a sampling duration of 1 minute and a duration of 1.28 seconds per sample. The dataset, in CSV format, includes vibration signals for analyzing bearing fault types and characteristics. Fault diagnosis studies based on this dataset employ various algorithms such as standard deviation, FFT spectrum, and envelope spectrum for abnormal detection and fault classification. For instance, envelope spectrum analysis of acceleration signals aids in identifying outer race faults. This dataset not only fuels research in prediction and health management but also facilitates the practical application of intelligent operations and maintenance in the industry.

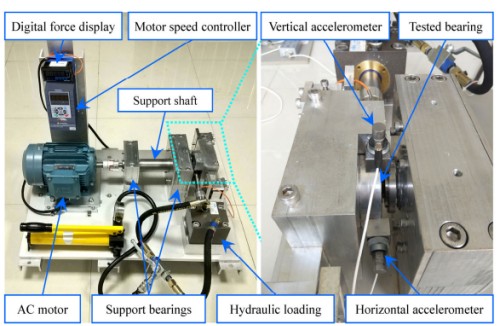

Fig. 4: Experimental platform of the XJTU-SY bearing dataset.

TABLE I: FAR Values for TEP.

| | PCA | | NMF | | SNMF | | ONMF | | GLNMFA | |
|---|---|---|---|---|---|---|---|---|---|---|
| | $T^2$ | SPE | $T^2$ | SPE | $T^2$ | SPE | $T^2$ | SPE | $T^2$ | SPE |
| IDV(1) | **0.00%** | 10.62% | 9.38% | **0.00%** | 16.83% | 0.63% | 46.25% | 0.63% | 0.40% | **0.00%** |
| IDV(2) | 0.63% | 18.13% | **2.50%** | 0.00% | 5.00% | 4.73% | 49.38% | **0.00%** | 3.00% | **0.00%** |
| IDV(3) | 0.00% | 13.75% | 3.13% | **0.00%** | 27.25% | **0.00%** | 48.25% | 0.20% | 0.02% | **0.00%** |
| IDV(4) | 5.63% | **0.00%** | 13.88% | 3.75% | 14.63% | 3.13% | 48.13% | **0.00%** | **0.20%** | 0.20% |
| IDV(5) | 3.75% | **0.00%** | 2.50% | 6.88% | 28.23% | 0.63% | 35.00% | 1.25% | **0.20%** | 1.00% |
| IDV(6) | 22.75% | 17.33% | 6.40% | 6.40% | **0.00%** | **0.00%** | 32.00% | **0.00%** | **0.00%** | **0.00%** |
| IDV(7) | 12.58% | 25.10% | 6.40% | 2.32% | 11.40% | **0.00%** | 12.00% | 0.60% | **0.00%** | 2.40% |
| IDV(8) | 82.51% | 1.25% | 4.16% | **0.00%** | 15.00% | **0.00%** | 6.80% | **0.00%** | **0.00%** | **0.00%** |
| IDV(9) | 4.83% | 22.50% | 6.40% | **0.56%** | 25.80% | 0.60% | 15.40% | **0.00%** | 0.40% | 2.40% |
| IDV(10) | 4.38% | 22.50% | 6.28% | **0.28%** | 31.20% | 0.60% | 21.80% | 2.20% | **0.40%** | 0.40% |
| Average | 13.71% | 13.12% | 6.10% | 2.02% | 17.53% | 1.03% | 31.50% | 0.49% | **0.46%** | **0.64%** |

TABLE II: MAR Values for TEP.

| | PCA | | NMF | | SNMF | | ONMF | | GLNMFA | |
|---|---|---|---|---|---|---|---|---|---|---|
| | $T^2$ | SPE | $T^2$ | SPE | $T^2$ | SPE | $T^2$ | SPE | $T^2$ | SPE |
| IDV(1) | 0.50% | 0.25% | 1.25% | 1.25% | 8.38% | 1.02% | 2.02% | 4.76% | **0.17%** | **0.00%** |
| IDV(2) | **0.25%** | 16.87% | 0.38% | 5.38% | 6.88% | **0.35%** | 30.13% | 84.13% | 0.98% | 0.95% |
| IDV(3) | 1.38% | 16.25% | 1.75% | 2.25% | 27.38% | 4.88% | 27.75% | 2.88% | **0.20%** | **0.19%** |
| IDV(4) | 1.57% | 14.58% | 1.00% | 3.50% | 12.63% | 7.63% | 47.25% | 9.88% | **0.20%** | **0.19%** |
| IDV(5) | 1.00% | 12.35% | 1.75% | 3.00% | 24.88% | 3.00% | 22.13% | 3.75% | **0.20%** | **0.19%** |
| IDV(6) | 27.25% | 7.25% | **0.00%** | 0.02% | 42.00% | 42.13% | 0.04% | 0.11% | 0.20% | **0.14%** |
| IDV(7) | 0.50% | 0.36% | **0.01%** | **0.04%** | 0.09% | 0.20% | 0.10% | 0.20% | 0.20% | 0.20% |
| IDV(8) | 1.50% | 7.13% | **0.00%** | **0.04%** | 0.03% | 0.20% | 0.04% | 0.20% | 0.20% | 0.20% |
| IDV(9) | 98.75% | 93.13% | 97.16% | **92.52%** | 97.25% | 98.31% | 95.21% | 97.33% | **94.27%** | 96.20% |
| IDV(10) | 73.5% | 77.13% | **0.00%** | **0.04%** | **0.00%** | 0.20% | 0.09% | 0.19% | 0.20% | 0.20% |
| Average | 20.62% | 24.53% | 10.33% | 10.08% | 21.95% | 15.79% | 22.47% | 20.34% | **9.68%** | **9.85%** |

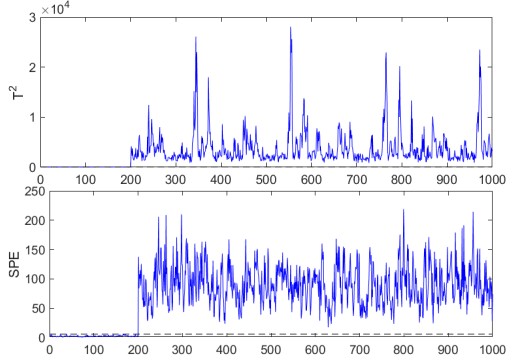

Fig. 5: Bearing 3-1 detection results in the XJTU-SY used PCA.

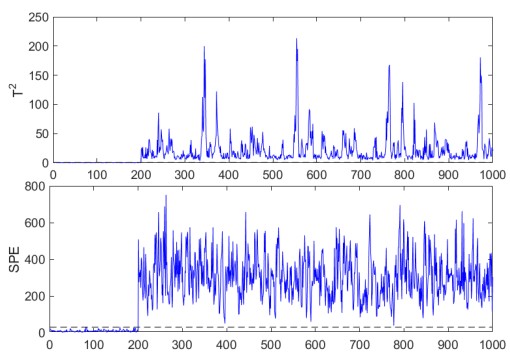

Fig. 6: Bearing 3-1 detection results in the XJTU-SY used NMF.

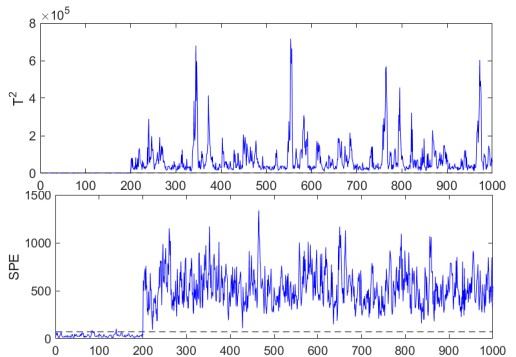

Fig. 7: Bearing 3-1 detection results in the XJTU-SY used SNMF.

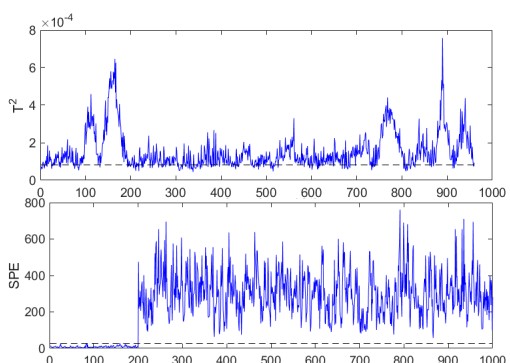

Fig. 8: Bearing 3-1 detection results in the XJTU-SY used ONMF.

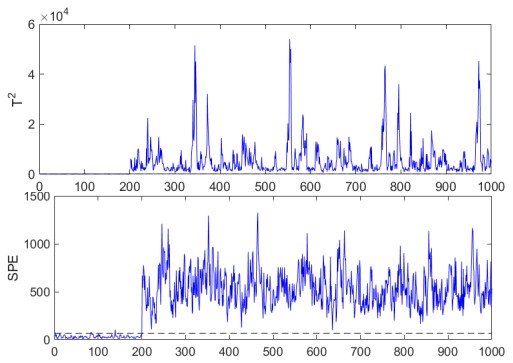

Fig. 9: Bearing 3-1 detection results in the XJTU-SY used GLNMFA.

The pictures Fig. 5 to Fig. 9 show detection performance of Bearing 3-1 for XJTU-SY bearing. In this dataset, faults are artificially introduced starting at the 201st data point. The figure illustrates the fault points on the horizontal axis, the calculated sample values on the vertical axis, and the dashed line indicates the control limit. A fault point exceeding the control limit confirms the occurrence of a fault, while a point below the limit suggests that no fault has occurred or the fault has not been accurately detected.

As can be seen from Table III and Table IV, although the detection result of GLNMFA is not optimal at individual fault points (for example, MAR Values for Bearing 2-5), considering the average results, the detection efficiency of GLNMFA for XJTU-SY data set is nonetheless remains.

## V. Conclusions

In this paper, a novel non-negative matrix decomposition model GLNMFA is proposed for fault detection in industrial systems. The model improves the fault detection efficiency by adding regularization terms to the classical NMF model. The graph regularization term plays a pivotal role in preserving the inherent graph structure characteristics of the data, which is essential for comprehending the interdependencies and influences among various components within a complex system. The incorporation of nuclear norm regularization terms results in a sparser representation, aiding in the elimination of noise and insignificant features, thus enhancing the model's interpretability. Experimental validation on the TEP and XJTU-SY datasets has substantiated the effectiveness of the proposed NMF algorithm in fault detection. When compared to existing fault detection methods, this model exhibits superior performance in terms of detection accuracy, robustness, and computational efficiency.

Despite the positive results of this study, there is still room for further improvement. Future work can focus on: parameter selection and tuning strategies to automatically determine optimal regularization parameters. Algorithm performance optimization in real-time fault detection scenarios. Multi-modal data fusion to further improve the accuracy of fault detection.

## VI. Acknowledgements

This research was funded by the Open Fund of the Key Laboratory of Cyber-Physical Fusion Intelligent Computing (South-Central Minzu University), State Ethnic Affairs Commission under Grant CPFIC202303.

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

TABLE III: FAR Values for XJTU-SY.

| | PCA | | NMF | | SNMF | | ONMF | | GLNMFA | |
|---|---|---|---|---|---|---|---|---|---|---|
| | $T^2$ | SPE | $T^2$ | SPE | $T^2$ | SPE | $T^2$ | SPE | $T^2$ | SPE |
| Bearing 3-1 | 1.26% | 0.22% | 3.13% | 0.63% | 10.62% | 1.88% | 52.29% | 0.63% | **0.31%** | **0.17%** |
| Bearing 1-4 | 99.36% | 99.36% | 6.40% | **0.04%** | 29.20% | 2.60% | 25.40% | 1.80% | **0.40%** | 1.60% |
| Bearing 2-5 | 0.63% | **0.00%** | 6.08% | 0.24% | 27.40% | **0.00%** | 24.60% | 1.40% | **0.40%** | 2.20% |
| Average | 33.75% | 33.19% | 5.20% | 0.30% | 22.41% | 1.49% | 34.10% | **1.28%** | **0.37%** | 1.32% |

TABLE IV: MAR Values for XJTU-SY.

| | PCA | | NMF | | SNMF | | ONMF | | GLNMFA | |
|---|---|---|---|---|---|---|---|---|---|---|
| | $T^2$ | SPE | $T^2$ | SPE | $T^2$ | SPE | $T^2$ | SPE | $T^2$ | SPE |
| Bearing 3-1 | 18.23% | **0.00%** | 6.55% | 4.76% | 10.12% | 4.64% | 1.55% | 4.76% | **0.06%** | 0.14% |
| Bearing 1-4 | 75.95% | 16.90% | **0.00%** | 0.04% | 0.02% | **0.02%** | 0.05% | 0.20% | 0.20% | 0.20% |
| Bearing 2-5 | 4.76% | 4.76% | **0.03%** | **0.04%** | **0.03%** | 0.20% | 0.05% | 0.19% | 0.20% | 0.20% |
| Average | 32.98% | 7.22% | 2.19% | 1.61% | 3.39% | 1.62% | 0.55% | 1.72% | **0.15%** | **0.18%** |

[5] R. Isermann, "Supervision, fault-detection and fault-diagnosis methods—an introduction," *Control Engineering Practice*, vol. 5, no. 5, pp. 639–652, 1997.

[6] W. Qinghua, Z. Youyun, C. Lei, and Z. Yongsheng, "Fault diagnosis for diesel valve trains based on non-negative matrix factorization and neural network ensemble," *Mechanical Systems and Signal Processing*, vol. 23, no. 5, pp. 1683–1695, 2009.

[7] N. Md Nor, C. R. Che Hassan, and M. A. Hussain, "A review of data-driven fault detection and diagnosis methods: Applications in chemical process systems," *Reviews in Chemical Engineering*, vol. 36, no. 4, pp. 513–553, 2020.

[8] M. J. Zuo, J. Lin, and X. Fan, "Feature separation using ica for a one-dimensional time series and its application in fault detection," *Journal of Sound and Vibration*, vol. 287, no. 3, pp. 614–624, 2005.

[9] M. Tamura and S. Tsujita, "A study on the number of principal components and sensitivity of fault detection using pca," *Computers & Chemical Engineering*, vol. 31, no. 9, pp. 1035–1046, 2007.

[10] A. Simoglou, E. Martin, and A. Morris, "Canonical correlation analysis in process fault detection," *IFAC Proceedings Volumes*, vol. 33, no. 11, pp. 1011–1016, 2000.

[11] Y.-X. Wang and Y.-J. Zhang, "Nonnegative matrix factorization: A comprehensive review," *IEEE Transactions on Knowledge and Data Engineering*, vol. 25, no. 6, pp. 1336–1353, 2012.

[12] W. Guo, H. Che, and M.-F. Leung, "Tensor-based adaptive consensus graph learning for multi-view clustering," *IEEE Transactions on Consumer Electronics*, vol. 70, no. 2, pp. 4767–4784, 2024.

[13] J. Tang, X. Ceng, and B. Peng, "New methods of data clustering and classification based on nmf," in *2011 International Conference on Business Computing and Global Informatization*. IEEE, 2011, pp. 432–435.

[14] H. Kim and H. Park, "Sparse non-negative matrix factorizations via alternating non-negativity-constrained least squares for microarray data analysis," *Bioinformatics*, vol. 23, no. 12, pp. 1495–1502, 2007.

[15] M. T. Belachew, "Efficient algorithm for sparse symmetric nonnegative matrix factorization," *Pattern Recognition Letters*, vol. 125, pp. 735–741, 2019.

[16] W.-S. Zheng, S. Z. Li, J.-H. Lai, and S. Liao, "On constrained sparse matrix factorization," in *2007 IEEE 11th International Conference on Computer Vision*. IEEE, 2007, pp. 1–8.

[17] X. Xiu, J. Fan, Y. Yang, and W. Liu, "Fault detection using structured joint sparse nonnegative matrix factorization," *IEEE Transactions on Instrumentation and Measurement*, vol. 70, pp. 1–11, 2021.

[18] D. Cai, X. He, J. Han, and T. S. Huang, "Graph regularized nonnegative matrix factorization for data representation," *IEEE Transactions on Pattern Analysis and Machine Intelligence*, vol. 33, no. 8, pp. 1548–1560, 2010.

[19] S. Huang, H. Wang, T. Li, T. Li, and Z. Xu, "Robust graph regularized nonnegative matrix factorization for clustering," *Data Mining and Knowledge Discovery*, vol. 32, pp. 483–503, 2018.

[20] Y. Wang and R. Zhu, "Post-processing of graph regularized nonnegative matrix factorization algorithm for cancer gene clustering," in *2022 International Conference on Intelligent Manufacturing, Advanced Sensing and Big Data (IMASBD)*. IEEE, 2022, pp. 87–91.

[21] C. Ding, T. Li, W. Peng, and H. Park, "Orthogonal nonnegative matrix t-factorizations for clustering," in *Proceedings of the 12th ACM SIGKDD International Conference on Knowledge Discovery and Data Mining*, 2006, pp. 126–135.

[22] D. D. Lee and H. S. Seung, "Learning the parts of objects by non-negative matrix factorization," *Nature*, vol. 401, no. 6755, pp. 788–791, 1999.

[23] X.-b. Li, Y.-p. Yang, and W.-d. Zhang, "Fault detection method for non-gaussian processes based on non-negative matrix factorization," *Asia-Pacific Journal of Chemical Engineering*, vol. 8, no. 3, pp. 362–370, 2013.

[24] S. Węglarczyk, "Kernel density estimation and its application," in *ITM Web of Conferences*, vol. 23. EDP Sciences, 2018, p. 00037.

[25] Z. Chen, Z. O'Neill, J. Wen, O. Pradhan, T. Yang, X. Lu, G. Lin, S. Miyata, S. Lee, C. Shen *et al.*, "A review of data-driven fault detection and diagnostics for building hvac systems," *Applied Energy*, vol. 339, p. 121030, 2023.

[26] C. Li, H. Che, M.-F. Leung, C. Liu, and Z. Yan, "Robust multi-view non-negative matrix factorization with adaptive graph and diversity constraints," *Information Sciences*, vol. 634, pp. 587–607, 2023.

[27] W. Zhang, S. Yu, L. Wang, W. Guo, and M.-F. Leung, "Constrained symmetric non-negative matrix factorization with deep autoencoders for community detection," *Mathematics*, vol. 12, no. 10, p. 1554, 2024.

[28] L. Luo, J. Yang, J. Qian, and J. Yang, "Nuclear norm regularized sparse coding," in *2014 22nd International Conference on Pattern Recognition*. IEEE, 2014, pp. 1834–1839.

[29] Y. Cai, H. Che, B. Pan, M.-F. Leung, C. Liu, and S. Wen, "Projected cross-view learning for unbalanced incomplete multi-view clustering," *Information Fusion*, vol. 105, p. 102245, 2024.

[30] B. Pan, C. Li, H. Che, M.-F. Leung, and K. Yu, "Low-rank tensor regularized graph fuzzy learning for multi-view data processing," *IEEE Transactions on Consumer Electronics*, 2023.

[31] S. Chen and W. Guo, "Auto-encoders in deep learning—a review with new perspectives," *Mathematics*, vol. 11, no. 8, p. 1777, 2023.

[32] Y. Zhou, A. Amimeur, C. Jiang, D. Dou, R. Jin, and P. Wang, "Density-aware local siamese autoencoder network embedding with autoencoder graph clustering," in *2018 IEEE International Conference on Big Data (Big Data)*. IEEE, 2018, pp. 1162–1167.

[33] L. V. Utkin, A. V. Podolskaja, and V. S. Zaborovsky, "A robust interval autoencoder," in *2017 International Conference on Control, Artificial Intelligence, Robotics & Optimization (ICCAIRO)*. IEEE, 2017, pp. 115–120.

[34] Y. Dong, H. Che, M.-F. Leung, C. Liu, and Z. Yan, "Centric graph regularized log-norm sparse non-negative matrix factorization for multi-view clustering," *Signal Processing*, vol. 217, p. 109341, 2024.

[35] X. Yang, H. Che, M.-F. Leung, and C. Liu, "Adaptive graph nonnegative matrix factorization with the self-paced regularization," *Applied Intelligence*, vol. 53, no. 12, pp. 15 818–15 835, 2023.

[36] K. Chen, H. Che, X. Li, and M.-F. Leung, "Graph non-negative matrix factorization with alternative smoothed l 0 regularizations," *Neural Computing and Applications*, vol. 35, no. 14, pp. 9995–10 009, 2023.

[37] H. Che and J. Wang, "A nonnegative matrix factorization algorithm based on a discrete-time projection neural network," *Neural Networks*, vol. 103, pp. 63–71, 2018.

[38] S. Xiong, Y. Yang, F. Fekri, and J. C. Kerce, "Tilp: Differentiable learning of temporal logical rules on knowledge graphs," *ArXiv Preprint ArXiv:2402.12309*, 2024.

[39] S. Xiong, Y. Yang, A. Payani, J. C. Kerce, and F. Fekri, "Teilp: Time prediction over knowledge graphs via logical reasoning," in *Proceedings of the AAAI Conference on Artificial Intelligence*, vol. 38, no. 14, 2024, pp. 16 112–16 119.

[40] J. Harper and D. Wells, "Recent advances and future developments in pgd," *Prenatal Diagnosis*, vol. 19, no. 13, pp. 1193–1199, 1999.

[41] R. Merris, "Laplacian matrices of graphs: a survey," *Linear Algebra and Its Applications*, vol. 197, pp. 143–176, 1994.

[42] D. Hajinezhad, T.-H. Chang, X. Wang, Q. Shi, and M. Hong, "Nonnegative matrix factorization using admm: Algorithm and convergence analysis," in *2016 IEEE International Conference on Acoustics, Speech and Signal Processing (ICASSP)*. IEEE, 2016, pp. 4742–4746.

[43] D. Song, D. A. Meyer, and M. R. Min, "Fast nonnegative matrix factorization with rank-one admm," in *NIPS 2014 Workshop on Optimization for Machine Learning (OPT2014)*, 2014.

[44] D. L. Sun and C. Fevotte, "Alternating direction method of multipliers for non-negative matrix factorization with the beta-divergence," in *2014 IEEE international conference on acoustics, speech and signal processing (ICASSP)*. IEEE, 2014, pp. 6201–6205.

[45] Q. Wei, N. Dobigeon, and J.-Y. Tourneret, "Fast fusion of multi-band images based on solving a sylvester equation," *IEEE Transactions on Image Processing*, vol. 24, no. 11, pp. 4109–4121, 2015.

[46] R. A. Sadek, "Svd based image processing applications: state of the art, contributions and research challenges," *ArXiv Preprint arXiv:1211.7102*, 2012.

[47] C. Zhang, Q. Guo, and Y. Li, "Fault detection in the tennessee eastman benchmark process using principal component difference based on k-nearest neighbors," *IEEE Access*, vol. 8, pp. 49 999–50 009, 2020.

[48] F. Harrou, M. N. Nounou, H. N. Nounou, and M. Madakyaru, "Statistical fault detection using pca-based glr hypothesis testing," *Journal of Loss Prevention in the Process Industries*, vol. 26, no. 1, pp. 129–139, 2013.

[49] B. Wang, Y. Lei, N. Li, and N. Li, "A hybrid prognostics approach for estimating remaining useful life of rolling element bearings," *IEEE Transactions on Reliability*, vol. 69, no. 1, pp. 401–412, 2018.

[50] X. Zhang, X. Xiu, and C. Zhang, "Structured joint sparse orthogonal nonnegative matrix factorization for fault detection," *IEEE Transactions on Instrumentation and Measurement*, vol. 72, pp. 1–15, 2023.
