# OpenReview forum: "Graph Low-rank Non-negative Matrix Factorization with Auto-encoders for Fault Detection"
_IEEE.org/ICIST/2024/Conference — IEEE ICIST 2024 Conference Submission_

### Official Review · Reviewer_oiBc · 2024-08-22
**This article is well written and can be accepted.**

**Rating:** 7
**Confidence:** 3

**Review:**

1.There are many grammatical and typographical errors in the manuscript. Please check the full text carefully and correct them.
2.In simulation section, more analysis and descriptions should be given to show the effectiveness of the developed method.
3.To make it easier for readers to understand the novelties of this paper, it would be better for the authors to add the control frame diagram.
4.The engineering background of the proposed problem should be stated more clearly to readers. The literature review is insufficient. Some recently published papers should be included in the references list.

---

### Official Review · Reviewer_CWqX · 2024-08-27
**In this paper, a novel non-negative matrix decomposition model GLNMFA is proposed for fault detection in industrial systems.**

**Rating:** 7
**Confidence:** 3

**Review:**

a The language in the article needs further polishing.
b The references should be updated. Some closely related and new references should be added to show to further explain the novelty and innovation of the work.
c For the results presented in the Figures in the simulation, more explanations on them seem necessary and helpful to readers.

---

### Official Review · Reviewer_qgfW · 2024-08-28
**Graph Low-rank Non-negative Matrix Factorization with Auto-encoders for Fault Detection**

**Rating:** 7
**Confidence:** 2

**Review:**

In this paper, a novel non-negative matrix decomposition model GLNMFA is proposed for fault detection in industrial systems.
a The language in the article needs further polishing.
b The references should be updated. Some closely related and new references should be added to show to further explain the novelty and innovation of the work.
c For the results presented in the Figures in the simulation, more explanations on them seem necessary and helpful to readers.

---

### Decision · Program_Chairs · 2024-09-06

Accept (Oral)